# Genomewide Analysis and Biological Characterization of Cathelicidins with Potent Antimicrobial Activity and Low Cytotoxicity from Three Bat Species

**DOI:** 10.3390/antibiotics11080989

**Published:** 2022-07-22

**Authors:** Munjeong Choi, Hye-sun Cho, Byeongyong Ahn, Somasundaram Prathap, Soundrarajan Nagasundarapandian, Chankyu Park

**Affiliations:** Department of Stem Cell and Regenerative Biotechnology, Konkuk University, Hwayang-dong, Gwangjin-gu, Seoul 05029, Korea; alooomjc@gmail.com (M.C.); chssky77@gmail.com (H.-s.C.); ahn.b@outlook.com (B.A.); prathap@konkuk.ac.kr (S.P.); sundarmeets@gmail.com (S.N.)

**Keywords:** bat, cathelicidin, antimicrobial peptide, antimicrobial activity, antifungal activity

## Abstract

Cathelicidins are potent antimicrobial peptides with broad spectrum antimicrobial activity in many vertebrates and an important component of the innate immune system. However, our understanding of the genetic variations and biological characteristics of bat cathelicidins is limited. In this study, we performed genome-level analysis of the antimicrobial peptide cathelicidins from seven bat species in the six families, listed 19 cathelicidin-like sequences, and showed that the number of functional cathelicidin genes differed among bat species. Based on the identified biochemical characteristics of bat cathelicidins, three cathelicidins, HA-CATH (from *Hipposideros armiger*), ML-CATH (from *Myotis lucifugus*), and PD-CATH (from *Phyllostomus discolor*), with clear antimicrobial signatures were chemically synthesized and evaluated antimicrobial activity. HA-CATH showed narrow-spectrum antibacterial activity against a panel of 12 reference bacteria, comprising 6 Gram-negative and 6 Gram-positive strains. However, ML-CATH and PD-CATH showed potent antibacterial activity against a broad spectrum of Gram-negative and Gram-positive bacteria with minimum inhibitory concentration (MIC) of 1 and 3 μg/mL, respectively, against *Staphylococcus aureus.* ML-CATH and PD-CATH also showed antifungal activities against *Candida albicans* and *Cryptococcus cuniculi* with MIC of 5 to 40 μg/mL, respectively, and 80% inhibition of the metabolism of *Mucor hiemalis* hyphae at 80 μg/mL, while displaying minimal cytotoxicity to HaCaT cells. Taken together, although the spectrum and efficacy of bat cathelicidins were species-dependent, the antimicrobial activity of ML-CATH and PD-CATH was comparable to that of other highly active cathelicidins in vertebrates while having negligible cytotoxicity to mammalian cells. ML-CATH and PD-CATH can be exploited as promising candidates for the development of antimicrobial therapeutics.

## 1. Introduction

Chiroptera (bats) are the second largest order of mammals following rodents [1], and they possess unique biological features, such as laryngeal echolocation [2], vocal learning [3], the ability to fly [4], and exceptional longevity [5], that are the rarest among all mammals. Additionally, bats are known as reservoir hosts of many zoonotic viruses, including the Nipah virus [6], Ebola virus [7] and coronavirus [8]. With no observed pathological signs of the disease, they can better tolerate viral infections than most other mammals because of their adaptation to antiviral immune responses [9]. In particular, the similarity between severe acute respiratory syndrome-coronavirus-2 (SARS-CoV-2), the causative agent of current coronavirus disease 2019 outbreaks, and bat-borne coronavirus (Bat CoV RATG13) illuminated bats as special reservoirs for viral pathogens that cause severe diseases in other mammals [10]. Currently, various efforts to develop therapeutics against SARS-CoV-2 are being made [11,12].

Comparative analyses of the immune systems of bats and other mammals have been conducted, which suggest that bats have evolved novel mechanisms to limit proinflammatory responses induced by viral infections, while maintaining type I interferon (IFN) responses to restrict viral propagation [13,14,15]. Many severe viral infections cause excessive inflammation-associated pathogenesis in humans [16]. Studies have also shown the conservation of cytosolic RNA sensors, including pattern recognition receptors such as Toll-like receptors [14]. To better understand the genetic and evolutionary mechanisms underlying the unique adaptations of bats, studies on diverse bat genomes, including the Bat1K initiative, are being conducted [9,17,18]. However, the complex pathogenicity of viral infections in bats and its impact on bat fitness have not yet been clearly elucidated [19].

Antimicrobial peptides (AMPs), also known as host defense peptides, are important components of the innate immune system that are found in various organisms [20]. In addition to their broad range of antimicrobial activities against microbes, including bacteria, parasites, fungi, and virus [21,22,23,24,25], they act as immunomodulatory molecules [26,27,28]. Cathelicidins are potent AMPs consisting of an evolutionarily conserved cathelin-like domain (CLD) and an active mature peptide region at the C-terminus [29]. The release of mature peptides via enzymatic cleavage by specific proteases results in the generation of functional AMPs [30,31]. The expression levels of AMPs, including cathelicidins, are associated with the anti-inflammatory responses and clearance of pathogens [26,32,33].

Recent studies have revealed various functions other than antibacterial activity of vertebrate cathelicidins, particularly for LL-37 and CRAMP of humans and mice, respectively [34,35]. For example, a defective intestinal expression of CRAMP in non-obese diabetic (NOD) mice induced the colonic dysbiosis and could contribute to autoimmunity in the pancreas in turn [36]. Moreover, endogenous CRAMP and LL-37 are suggested as inactivators of Zika virus by inhibiting viral replication and directly inactivating viral virions, thus exhibiting therapeutic potential against the viral infection in vivo [37]. The protective role of LL-37 against *Candida auris* known for causing severe infectious disease with high mortality attributed to multidrug-resistance against all major classes of antifungal drugs was also reported [38]. In addition, in vivo free-radical scavenging ability of a cathelicidin was reported from frogs [39].

The availability of genome sequences from diverse vertebrates provides a window of opportunity to identify genetic and functional diversity of cathelicidins [40,41,42]. However, understanding the biological effects of newly discovered cathelicidins from the high throughput analysis is slowly progressing. In this study, we conducted genome-wide in silico analysis of cathelicidin genes from seven bat species in the six families, *Rousettus aegyptiacus*, *Hipposideros armiger*, *Miniopterus natalensis*, *Myotis lucifugus*, *Rhinolophus ferrumequinum*, *Phyllostomus discolor*, and *Desmodus rotundus*, to maximize the genetic diversity and experimentally characterize their biological effects, including antibacterial, antifungal, and mammalian cytotoxic effects, in addition to the recent bioinformatics analysis of bat cathelicidins [43]. Our results contribute to a better understanding of the possible roles of AMPs in the antimicrobial defense of bats and can be exploited for the development of antimicrobial therapeutics.

## 2. Results

### 2.1. Identification of 27 AMP Family Genes via the In Silico Analysis of R. ferrumequinum Genome

To understand the genetic characteristics of the AMP subgenome in bats, we conducted in silico identification of classical AMP encoding genes from the reference genome (mRhiFer1_v1.p, GCF_004115265.1) of *R. ferrumequinum* as a representative species. We identified a total of 27 putative AMP genes from *R. ferrumequinum*, including three alpha-defensins, 21 beta-defensins, one cathelicidin, one hepcidin, and one LEAP-2, using National Center for Biotechnology Information (NCBI) genome blast with query coverage > 50% and e-value < 0.001 (Appendix A). The number of AMP genes identified from our analysis was consistent with the gene numbers reported in *R. ferrumequinum* for each AMP family in the NCBI protein database without finding additional AMP genes.

### 2.2. Differences in the Number of Functional Cathelicidin Genes among Seven Bat Species Belonging to Different Families

To determine the number of functional cathelicidin genes in bat genomes, 64 unique cathelicidin protein sequences from 36 vertebrate species were blasted using the tblastn mode against the genomes of seven different bat species, *R. aegyptiacus, H. armiger, M. natalensis, M. lucifugus, R. ferrumequinum, P. discolor and D. rotundus*. Consequently, 12 cathelicidin-like sequences were identified in the analysis with query coverage > 50% and e-value < 0.001 (Appendix A). In addition, BLASTP analysis using the same queries against nonredundant protein sequences of each bat species (taxid:9407, 186990, 291302, 59463, 59479, 89673, and 9430) resulted in the identification of 19 matches of cathelicidin-like sequences (Appendix A). Further analysis showed that the difference in the number of matches was due to the discrepancy in the annotation of cathelicidin genes between the current genome assemblies and available sequences of bat cathelicidins in the NCBI protein database (Appendix A). When the cathelicidin sequences of bats in public databases were taxonomically classified, the number of cathelicidin genes varied across bat species. A single cathelicidin-like gene has been detected in the genomes of *H. armiger* (LOC109375616) and *M. natalensis* (LOC107529421). Two putative cathelicidin isoforms, XP_036075573.1 and KAF6473429.1, have been reported in *R. aegyptiacus*, but they seem to result from two different annotations of an identical gene. In *D. rotundus,* the antimicrobial activity-containing region was predicted only from one (XP_024421797.1) of the two cathelicidin-like sequences (XP_024421797.1 and XP_024421798.1). Similarly, two cathelicidin-like sequences, KAF6312594.1 and XP_032988513.1, were encoded from an identical locus in *R. ferrumequinum,* and the structure of KAF6312595.1 lacks an exon. Therefore, five out of the seven species seem to have a single functional cathelicidin gene in their genomes, and more than one cathelicidin could be present in *M. lucifugus* and *P. discolor* considering the presence of multiple sequences with antimicrobial signatures, as described in Appendix A.

### 2.3. In Silico Determination of the Antimicrobial Activity Core Regions of Three Bat Cathelicidins

We conducted an in silico prediction of the antimicrobial activity core region for the 19 bat cathelicidin-like sequences to define the signal peptide, conserved CLD domain, and mature peptide region using two different prediction tools, antimicrobial sequence scanning system (AMPA) and database of antimicrobial activity and structure of peptides (DBAASP) (Appendix A). Consequently, five putatively functional sequences, KAF6473429.1, XP_019486615.1, XP_006108360.2, XP_035886276.1, and XP_028374415.1, with the complete pre-pro protein structure of cathelicidins and predicted presence of core antimicrobial activity regions were determined (Appendix A). Among them, three predicted antimicrobial activity core regions corresponding to the cathelicidins of *H. armiger* (XP_019486615.1), *M. lucifugus* (XP_006108360.2) and *P. discolor* (XP_028374415.1) which were named as ΔHA-CATH, ΔML-CATH, and ΔPD-CATH where “Δ” indicate a deletion of several amino acids from the nascent mature peptide region, namely HA-CATH, ML-CATH, and PD-CATH, respectively, were selected and subjected to the analysis of biochemical features using Antimicrobial Peptide Database (APD3). The three antimicrobial activity core regions or short form peptide regions (ΔHA-CATH, ΔML-CATH, and ΔPD-CATH) consist of 18 to 25 amino acids with 2.08 to 2.93 kDa in molecular weight, whereas the entire mature or long form peptides, HA-CATH, ML-CATH, and PD-CATH, were composed of 35 to 40 residues with 3.98 to 4.64 kDa in molecular weight (Table 1). The ratios of hydrophobic residues and net charges for the short- and long-form peptides were 31 to 50% and +3 to +6, and 30 to 40% and +4 to +8, respectively. Their amino acid sequence similarity to known cathelicidins was highest with cathelicidins from sheep (SMAP-29), dogs (K9CATH), frogs (Temporin-CPb, Palustrin-2CG1), and fish (TP4, HKPLP) and the level ranged from 38.64 to 51.22%.

### 2.4. Confirmation of the Antibacterial Activity of Three Bat Cathelicidins

Six bat-cathelicidin peptides (ΔHA-CATH, ΔML-CATH, ΔPD-CATH, HA-CATH, ML-CATH, and PD-CATH) were synthesized, and antibacterial activities were evaluated against a panel of 12 reference bacteria, comprised of 6 Gram-negative strains, *Escherichia coli*, *Pseudomonas aeruginosa*, *Salmonella enterica serovar* Typhimurium, *Acinetobacter baumannii*, *Klebsiella pneumoniae* subsp. *pneumoniae*, and *Enterobacter cloacae* subsp. *cloacae*, and 6 Gram-positive strains, *Staphylococcus aureus*, *Bacillus cereus*, *Enterococcus faecalis*, *Streptococcus agalactiae*, *Streptococcus dysgalactiae*, and *Streptococcus equi* subsp. *zooepidemicus*, which are associated with human opportunistic and/or nosocomial pathogens (Table 2). ML-CATH and PD-CATH showed highly potent antibacterial activity against most strains in our bacterial panel with minimum inhibitory concentration (MIC) of 1–35 μg/mL (0.3–8.8 μM) and 3–30 μg/mL (0.7–7.3 μM) except *P. aeruginosa* (>40 μg/mL or 9.8 μM). In contrast, HA-CATH possessed a moderate antibacterial activity with MIC of 5–34 μg/mL (1.1–7.3 μM) against *E. coli*, *A. baumannii*, *K. pneumoniae* among Gram-negative strains and *S. aureus* and *B. cereus* among Gram-positive strains, indicating that the antimicrobial potency and spectrum of bat cathelicidins varied among species. However, ΔHA-CATH, ΔML-CATH, and ΔPD-CATH, the short-form peptides corresponding to the predicted antibacterial core region, did not show any observable antibacterial activity against all strains of the reference bacterial panel, even at a high concentration (>40 μg/mL or 13.7 μM), indicating the prediction of active core region for the bat cathelicidins was not accurate.

### 2.5. Confirmation of the Antifungal Activity of Three Bat Cathelicidins

Six bat-cathelicidin peptides (ΔHA-CATH, ΔML-CATH, ΔPD-CATH, HA-CATH, ML-CATH, and PD-CATH) were evaluated for the presence of antifungal activity against three different fungal species, *Candida albicans (C. albicans)*, *Cryptococcus cuniculi (C. cuniculi)*, and *Mucor hiemalis (M. hiemalis)* (Table 3; Figure 1). The antifungal effect of the yeast-type fungi, *C. albicans* and *C. cuniculi,* was estimated in a manner similar to the antibacterial activity method. However, the antifungal activity of *M. hiemalis,* a mold-type fungus, was estimated based on the inhibition of the metabolic rate against hyphae. The treatment of ML-CATH and PD-CATH showed effective antifungal activities against the two yeast forms fungi with MIC of 5 to 40 μg/mL (1.2 to 10.1 μM) and *M. hiemalis* with a similar rate of metabolic inhibition (~80%) to fluvastatin sodium. In addition, intermediate levels of antifungal activity were observed against *C. albicans* and *C. cuniculi* from the treatment of ΔPD-CATH which did not show antibacterial activity against our reference bacterial panel, indicating the inherently strong antimicrobial potency of PD-CATH compared to the other bat cathelicidins tested in this study and differences in the effect of ΔPD-CATH against bacterial and fungal cell walls. In addition, HA-CATH only showed slight antifungal activity (MIC = 45 μg/mL or 9.7 μM) against *C. cuniculi*, indicating limited activity against fungi which is consistent with the antibacterial activity of HA-CATH (Table 2). The antifungal activity of the tested peptides seems to be higher for yeast than mold-type fungi. Consistent with the results of antibacterial analysis, ΔHA-CATH, ΔML-CATH, and ΔPD-CATH did not show antifungal effects against the reference strains, except ΔPD-CATH against *C. cuniculi*.

### 2.6. Negligible Cytotoxicity of Bat Cathelicidins against Human Keratinocytes

The level of cytotoxicity is important for the therapeutic use of AMPs as antimicrobial agents. To estimate the cytotoxicity of HA-CATH, ML-CATH, and PD-CATH in mammalian cells, the viability of HaCaT cells was evaluated for each peptide (Table 4). The results showed that no cytotoxicity was observed at 64 μg/mL concentration, and only less than 3% of cells were affected at 160 μg/mL for both HA-CATH and ML-CATH. For PD-CATH, the cytotoxicity was not observed at 64 μg/mL, but the cell viability was decreased to 10.4% at 160 μg/mL concentration, showing significant cytotoxicity at the high concentration. Therefore, the cytotoxicity on mammalian cells showed a positive correlation with the potency of bat cathelicidins. In addition, melittin, an AMP known to have a potent cell lytic activity, showed severe cell lysis, with 7.2% cell viability at 64 μg/mL concentration. These results indicate that the cytotoxic effect of HA-CATH and ML-CATH is at a minimal level, and given the effective bactericidal concentration of PD-CATH (3–30 μg/mL), the cytotoxicity is within the acceptable range for therapeutic use.

## 3. Discussion

Fifty genome sequences of diverse bat species with varying degrees of coverage are currently available in the public database (https://www.ncbi.nlm.nih.gov/genome, accessed on 22 June 2022). Considering the extreme phylogenetic diversity of bats, comparative analysis of their AMP genes may reveal interesting outcomes regarding the genetic and functional diversity of these molecules. The number of classical AMP genes of *R. ferrumequinum* was similar to that of naked mole rats when compared to other mammals (Appendix A) [25,44,45,46,47,48]. However, only limited information is available on the genetic and biological characteristics of bat AMPs, including cathelicidins. In addition, the results of our analysis to identify cathelicidin orthologs from the seven selected bat species showed few differences in annotations compared to the currently predicted protein sequences of cathelicidins in NCBI (Appendix A), suggesting the need of more cDNA information, especially for isoform-like sequences. Cathelicidins are probably the most potent AMPs in mammals, with variations in their activities in cases of multiple cathelicidins within a species [49]. Although few studies reported bioinformatic prediction of bat cathelicidins based on available sequencing information and experimental characterization of their antimicrobial activity [43,50], the number of tested bacterial strains and peptides was highly limited, and other characteristics such as fungicidal and metabolically inhibitory activities against fungi and cytotoxicity have not been validated experimentally. To the best of our knowledge, this is the first report of the antimicrobial activity of bat cathelicidins against diverse bacteria and fungi associated with human health and food pathogens.

A previous study briefly reported the bactericidal effect of a bat cathelicidin, Ml-LN-35, which is identical to ML-CATH, against *E. coli* and *B. cereus* [50]. The results were consistent with the results in this study with MICs ranging from 0.5 to 0.8 μM (Table 2). In contrast, antibacterial activity against *S. aureus* ATCC 6538 was not detected in the previous study, while the peptide showed highly potent anti-staphylococcal activity (1 μg/mL or 0.3 μM) as shown in Table 2. This inconsistency might be attributed to different experimental condition including media composition and peptide synthesis with some modification such as acetylation and amidation in the N- and C-terminal end, respectively, which can affect the nature and structure of AMPs and influence biological activity and stability of the molecules in turn [51,52,53].

In our functional analysis of bat cathelicidins, all three mature peptide region-derived peptides showed antibacterial activity against diverse bacterial strains. However, the truncated peptides corresponding to the predicted antimicrobial activity core region as a minimal unit for biological activity did not show substantial antibacterial activity, although the two different forms of peptides shared common characteristics in hydrophobicity and positive net charge (Table 1). This is somewhat different from the results of our previous studies to characterize cathelicidins of non-bat species [25,47,54].

The alteration of primary sequences and/or physicochemical properties of α-helical cationic AMPs, especially cathelicidins, could affect the formation of amphipathic α-helical structures, hydrophobicity, the density and distribution of charge locally or overall, and minimal length, influencing biological activities, including antimicrobial and mammalian cytotoxicity [55,56,57,58]. For example, a previous study showed that IG-13, corresponding to the minimal active domain of LL-37, a well-known human cathelicidin, was approximately 30-fold less active than LL-37 [58]. The smallest fragment, LL-19, did not form a clear α-helix in the buffer, although the peptide region originated from the α-helical region [55]. In addition, the hydrophobic tail length of the peptides could affect their ability to insert and penetrate the bacterial membrane [57]. Therefore, the loss of antimicrobial activity in the predicted antimicrobial activity core region peptides 18 to 25 amino acids in size of bat cathelicidins could be influenced by peptide length. Further studies are necessary to determine the amino acids critical for maintaining the antimicrobial activity of nascent mature peptides.

Among the mature peptide regions of the three cathelicidins, the specificity and strength of the antibacterial activities of ML-CATH and PD-CATH were highly similar (Table 2). This could be attributed to the similarities in peptide length, hydrophobicity, and net charge between the two peptides despite their sequence differences (Table 1; Appendix A).

In addition to the extensive functional studies on LL-37 and CRAMP, recent interests in AMPs as potent antibiotic agents lead to the identification and characterization of novel cathelicidins from diverse species including *Python bivittatus* (Burmese python), *Heterocephalus glaber* (naked mole rat), *Monodelphis domestica* (gray short-tailed opossum), *Lates calcarifer* (Asian sea bass) and *Sarcophilus harrisii* (Tasmanian devil) [25,47,54,59,60]. These studies suggested that cathelicidins are promising candidates to develop antimicrobial therapeutics considering potent broad-spectrum bactericidal activity, minimal cytotoxicity to mammalian cells, and inducibility of lower antibiotic resistance compared to classical antibiotics. Interestingly, bat cathelicidins, ML-CATH and PD-CATH, showed much broader and stronger antibacterial activities than all the peptides described above (Table 2). Furthermore, ML-CATH and PD-CATH showed milder toxicity compared to ModoCath1 which is one of the cathelicidins showing broad spectrum with high potency. The viability of human keratinocyte was affected at a five-fold higher concentration (160 µg/mL) in PD-CATH comparing to ModoCath1 (32 µg/mL) (Table 4). These characteristics could make ML-CATH and PD-CATH more suitable for pharmaceutical applications than other cathelicidins. However, further studies are still required to overcome intrinsic disadvantages of AMPs including high production cost, low stability in vivo, and deficiency of proper delivery system. An effective production system of recombinant cathelicidins using an engineered green fluorescent protein (GFP) as a tag protein was attempted for large-scale production of AMPs [61]. The conjugation of AMPs with nanoparticles has been suggested to overcome poor stability of AMPs in biological fluids and proteolytic degradation and to control pharmacokinetics [62,63,64].

In mammals, the copy number of cathelicidin genes varies among species, and a single functional cathelicidin gene is present in dogs, mice, and humans, whereas more than one functional gene is found in pigs, goats, sheep, cattle, and whales [65]. Intraspecies copy number variation has also been reported in pigs [48]. It is interesting to note that the antimicrobial potency of ML-CATH and PD-CATH, obtained from multiple cathelicidin gene-carrying bat species, was higher and more broadly effective against diverse bacterial strains than HA-CATH, obtained from species carrying only a single cathelicidin gene. The expansion of cathelicidin genes in certain bat species may be due to the functional importance of the molecule for their survival. It may also enhance the antimicrobial potency of paralogous genes in parallel.

The variations in the number of cathelicidin genes among different bat species can be attributed to the large phylogenetic diversity of bats in the order Chiroptera [66]. Moreover, in our analysis of the seven bat species, all annotated cathelicidin genes were mapped to the *CDC25A* and *NME6* interval, the evolutionarily conserved cathelicidin residing region in mammals [48], except *M. lucifugus* with less complete genome information, suggesting the evolutionary conservation of synteny in bats, similar to other mammals. Therefore, the copy number of cathelicidins in bats could vary from a single copy to multiple copies depending on the species, which is consistent with a previous analysis [43]. However, further refinement of the current bat genome assemblies and annotations is needed to determine the copy number of cathelicidin family genes more accurately.

Endogenous AMPs also showed antiviral activity against diverse viruses, including the human influenza virus, human immunodeficiency virus, rabies virus, West Nile virus (WNV), and herpes simplex virus (HSV) [25,67,68,69,70]. Among them, gramicidin S and melittin from *Bacillus brevis* and bee venom, respectively, showed noteworthy anti-SARS-CoV-2 effects with viral particle reduction compared to remdesivir, a repurposed drug used to treat the infection [71]. Antiviral activities against HSV and WNV were also reported for other cathelicidins, such as LL-37, Pb-CATH4, Hg-CATH, and ModoCath5 [25,70,72]. Because of the therapeutic potential of these peptides, it will be interesting to evaluate the antiviral activity of bat cathelicidins, which also showed negligible cytotoxicity to mammalian cells in this study. The antibacterial and antifungal activities of HA-CATH, ML-CATH, and PD-CATH observed in this study can serve as references for the potency and antimicrobial spectrum analyses of bat cathelicidins.

## 4. Materials and Methods

### 4.1. In Silico Identification of Cathelicidin-like Sequences in the Genome of Seven Bat Species

Four hundred sequences belonging to five major AMP families of vertebrates, alpha-defensins, beta-defensins, cathelicidin, hepcidin, and LEAP-2, were obtained similarly to the previous studies [25,47]. Briefly, the sequences of 2477 nonredundant AMPs were downloaded from UniProtKB/Swiss-Prot (http://www.uniprot.org/uniprot/, accessed on 16 March 2022; [73]) using the query “antimicrobial peptide AND reviewed: yes”. After removing sequences originated from nonvertebrates, the sequences corresponding to five major AMP families described above were collected (Appendix A) [25]. The sequences were blasted using the NCBI blastp mode (https://www.ncbi.nlm.nih.gov/, accessed on 16 March 2022) against two different reference genomes of *R. ferrumequinum* (mRhiFer1_v1.p, GCF_004115265.1 and mRhiFer1.p, GCA_014108255.1) with the genome coverage of 52.8 x and 73.3 x, respectively, in NCBI. In addition, 64 cathelicidin-like sequences from 400 vertebrate AMPs with both the CLD and mature domains were subjected to NCBI blastp and tblastn analyses against the reference genomes of additional six bat species with the coverage of 7 x to 218.6 x in NCBI genome database (https://www.ncbi.nlm.nih.gov/genome, accessed on 18 March 2022), including *R. aegyptiacus* (mRouAeg1.p, GCF_014176215.1), *H. armiger* (ASM189008v1, GCF_001890085.1), *M. natalensis* (Mnat.v1, GCF_001595765.1), *M. lucifugus* (Myoluc2.0, GCF_014108235.1), *P. discolor* (mPhyDis1.pri.v3, GCF_004126475.2, and mPhyDis1_v1.p, GCF_004126475.1), and *D. rotundus* (ASM294091v2, GCF_002940915.1). The prediction of genetic structures and positional confirmation of putative cathelicidin sequences in bat genomes were conducted using the NCBI (https://www.ncbi.nlm.nih.gov/genome/gdv, accessed on 28 March 2022) and the Bat1K genome browser (https://genome-public.pks.mpg.de, accessed on 1 April 2022). The last accession of genome databases was 1 May 2022.

### 4.2. In Silico Prediction of Putative Cathelicidins for Their Biological Activities

The in silico determination of the signal peptide and CLD regions was conducted using the SignalP 4.1 server (https://services.healthtech.dtu.dk/service.php?SignalP-4.1, accessed on 16 April 2022; [74]) and HMMER (https://www.ebi.ac.uk/Tools/hmmer/, accessed on 17 April 2022; [75]), respectively. AMPA (https://tcoffee.crg.eu/apps/ampa/do, accessed on 16 April 2022; [76]) and DBAASP (https://dbaasp.org/, accessed on 16 April 2022; [77]) were used to predict the potential antimicrobial activity domains using the default window size and threshold value. Protein secondary structures were predicted using the PSIPRED workbench (http://bioinf.cs.ucl.ac.uk/psipred/, accessed on 16 April 2022; [78]). The biochemical characteristics of predicted mature peptide sequences, including hydrophobicity, net charge, molecular weight, and similarity to known AMPs, were estimated using APD3 (https://aps.unmc.edu/, accessed on 1 May 2022; [79]).

### 4.3. Peptide Synthesis

Peptides corresponding to the predicted active core and mature peptide regions of cathelicidins were generated via solid-phase peptide synthesis, purified via high-performance liquid chromatography (HPLC), and confirmed by mass spectroscopy using a commercial service (GenScript, Piscataway Township, NJ, USA). Peptide names were given according to the acronym of binomial nomenclature of the species followed by “CATH” indicating cathelicidin. The peptide names and sequences for each species are ΔHA-CATH (N-LLRRGGRKIGQGLERIGQRIQGF-C) and HA-CATH (N-ILGRLRDLLRRGGRKIGQGLERIGQRIQGFFSNREPMEES-C) for *H. armiger,* ΔML-CATH (N-GIFILKHRRPIGRGIEIT-C) and ML-CATH (N-LNPLIKAGIFILKHRRPIGRGIEITGRGIKKFFSK-C) for *M. lucifugus*, and ΔPD-CATH (N-IAGRIAGKLIGDAINRHRERNRQRR-C) and PD-CATH (N-ILGPALRIGGRIAGRIAGKLIGDAINRHRERNRQRRG-C) for *P. discolor*. The synthesized peptides were dissolved in distilled water at a concentration of 4 mg/mL.

### 4.4. Evaluation of Antibacterial Activity

The antibacterial activities of the chemically synthesized peptides were evaluated against a panel of bacteria consisting of 6 Gram-negative strains, including *E. coli* ATCC 25922 (American Type Culture Collection, Manassas, VA, USA), *P. aeruginosa* ATCC 27853, *S.* Typhimurium ATCC 14028, *A. baumannii* KCTC (Korean Collection for Type Cultures, Jeongeup, Korea) 23254, *K. pneumoniae* KCTC 1726, and *E. cloacae* ATCC 13047, and 6 Gram-positive strains, including *S. aureus* ATCC 6538, *B. cereus* ATCC 10876, *E. faecalis* ATCC 29212, *S. agalactiae* ATCC 27956, *S. dysgalactiae* ATCC 27957, and *S. equi* ATCC 43079. Ampicillin (Sigma-Aldrich, St. Louis, MO, USA), chloramphenicol (Sigma-Aldrich), and gentamicin sulfate (Sigma-Aldrich) were used as controls for antibacterial activity. The MIC was determined by a colorimetric method using the Microbial Viability Assay Kit-WST (Dojindo, Kumamoto, Japan) according to the manufacturer’s manual and the Clinical and Laboratory Standards Institute (CLSI) guidelines (2018). Four colonies of each bacterium were inoculated into 5 mL of Luria-Bertani (LB) broth medium (BD Bioscience, Franklin Lakes, NJ, USA) or brain heart infusion broth (BHIB; BD Bioscience) at 37 °C for 6 h with shaking at 220 rpm. *E. faecalis* and streptococci were cultured in BHIB because of their slow growth, whereas all other bacteria were cultured in LB medium. The cells were washed by sterile saline (0.9% NaCl) twice and adjusted in a single well of a 96-well plate at a cell density of 1.5 × 10^5^ CFU/well in accordance with 0.5 McFarland standard. Subsequently, 180 µL/well of fresh Mueller-Hinton broth (MHB; BD Bioscience), except for E. faecalis and streptococci using BHIB was added to the plate using BHIB. Final concentrations of 1–40 µg/mL (0.3–19.3 μM) of each peptide and reference antibiotic were added to each well. The plates were then incubated at 37 °C for 6 h. Subsequently, 10 µL of the coloring reagent was added, and the mixture was incubated at 37 °C for 2 h. The absorbance of each well was measured at 450 nm using a microplate spectrophotometer (Bio-Rad, xMark spectrophotometer, Hercules, CA, USA). MIC values were determined when the difference in the absorbance values between treatments and blanks (media and coloring reagent only) was less than 0.05. Experiments were performed in triplicate.

### 4.5. Evaluation of Antifungal Activity

The antifungal activities of the peptides were evaluated against two yeast fungi, *C. albicans* KCTC 7270 and *C. cuniculi* KCTC 17232, and a mold fungus, *M. hiemalis* KCTC 26779. *C. albicans* and *C. cuniculi* were cultured in yeast mold broth (YMB; BD Science) agar at 35 °C and 25 °C, respectively. Cells were prepared in YMB for *C. albicans* and RPMI 1640 medium (with l-glutamine, without phenol red and sodium bicarbonate, Sigma Aldrich, Saint Louis, MO, USA) and adjusted to pH 7.0 with 0.165 M 3-(N-morpholino) propanesulfonic acid (MOPS) (iNtRON Biotechnology, Seoul, Korea), as previously described [80] for *C. cuniculi*. According to the 0.5 McFarland standard, cells at 1.5 × 10^5^ CFU/well were seeded into a 96-well plate containing 180 µL of fresh medium. Subsequently, peptides at final concentrations of 5–45 ug/mL (1.2–19.3 μM) peptides were added to each well. Ciclopirox (≥98% (HPLC), Sigma Aldrich) was used as a reference for the antifungal activity. The antifungal assay was conducted in the same manner as the antibacterial assay described above, according to the manufacturer’s protocol. Briefly, *C. albicans* and *C. cuniculi* were incubated at 35 °C for 6 h and 25 °C for 47 h, respectively, and 10 µL of the coloring reagent was added and further incubated for 18 h and 25 h. The MIC values were determined in the same way as for the antibacterial assay.

To evaluate the antifungal activity of peptides against *M. hiemalis*, the metabolic activity of hyphae was measured using the same colorimetric assay as described above, with slight modifications to a previous study [81]. Briefly, *M. hiemalis* was streaked onto malt extract broth (BD Biosciences) agar in a Petri dish and cultured at 25 °C for seven days. Then, conidia were collected from the dish by adding phosphate buffered saline (PBS) containing 0.05% Tween-80 (Sigma Aldrich) and filtered through sterilized 40 um cell strainer (SPL Life Science, Pocheon, Korea) to remove hyphae. After washing twice with PBS and centrifugation, 5 × 10^4^ conidia were counted using a hemocytometer, suspended in 50 µL of RPMI 1640 as described above, and seeded in wells of a 96-well plate. After 16 h of incubation at 25 °C for hyphae formation, the plate was centrifuged at 3000× *g* and the media were changed. The peptides were then added at final concentrations of 20, 40, and 80 µg/mL. Fluvastatin sodium (70 µM; Sigma-Aldrich) was used as a reference for antifungal activity. For the coloring reaction, 10 µL of the coloring solution (Microbial Viability Assay Kit-WST, Dojindo) was added to each well and incubated for 2 h. Subsequently, the plate was centrifuged, and the supernatant was transferred to a new 96-well plate. The absorbance at 450 nm (for the treated group and control) and 650 nm (for the background) was measured for each well using a microplate reader (xMarkTM spectrophotometer; Bio-Rad). The metabolic activity of the fungal cells was calculated using the following equation. All experiments were performed in triplicate.
Fungal metabolic activity (%) = 100 × (OD (treated group) − OD (background))/(OD (nevative control) − OD (Background))(1)

### 4.6. In Vitro Cytotoxicity Assay

HaCaT cells were cultured in Dulbecco’s modified Eagle’s medium (DMEM; Hyclone^TM^, Logan, UT, USA) with 10% fetal bovine serum (FBS; Hyclone^TM^) and 1% penicillin/streptomycin (Pen-Strep; Hyclone^TM^) up to 80% confluence at 37 ℃ and 5% CO_2_. Cells were detached from the plate by adding 0.25% trypsin-EDTA solution (Gibco^TM^; Carlsbad, CA, USA). In a 96-well plate, 2 × 10^4^ cells were seeded per well and cultured for 24 h. Subsequently, the medium was replaced with 100 μL DMEM (Hyclone^TM^) containing 10% heat-inactivated FBS. Peptides were added to each well at concentrations of 64 μg/mL and 160 μg/mL. As references for complete cell lysis, 64 μg/mL melittin (Sigma-Aldrich) and Triton X-100 (Sigma-Aldrich) were used. Untreated cells were used as negative controls. After 24 h incubation at 37 ℃ and 5 % CO_2_, 10 μL of coloring solution (Cell Proliferation Reagent WST-1^TM^; Sigma Aldrich) was added to each well according to the manufacturer’s instructions. The absorbance at 450 nm (treated group and control) and 650 nm (background) was measured in each well using a microplate reader (xMarkTM spectrophotometer; Bio-Rad). The cell viability was calculated using the following equation. All experiments were performed in triplicate.
Cell viability (%) = 100 × (OD (treated group) − OD(Background))/(OD (negative control) − OD (Background))(2)

### 4.7. Statistical Analysis

Statistical analysis was performed using Student’s paired *t*-test with a two-tailed distribution using RStudio (https://www.R-project.org/, accessed on 13 July 2022).

## 5. Conclusions

In summary, species-specific differences in genetic and functional characteristics of cathelicidins from seven bat species were studied. We listed 19 cathelicidin-like sequences from seven bat species and tested the biological activity of three cathelicidins (HA-CATH, ML-CATH, and PD-CATH). Among them, ML-CATH and PD-CATH showed potent antibacterial activity against a broad spectrum of Gram-negative and Gram-positive bacteria. The potency of antibacterial activity of ML-CATH and PD-CATH was comparable to that of other highly active cathelicidins reported in vertebrates. ML-CATH and PD-CATH also showed antifungal activities against *C. albicans, C. cuniculi*, and *M. hiemalis*. The cytotoxicity on HaCaT cells showed a positive correlation with the potency of bat cathelicidins, but given the effective bactericidal concentration, the cytotoxicity of bat cathelicidins is within the acceptable range for therapeutic exploitation. ML-CATH and PD-CATH are promising candidates for the development of peptide antibiotics, but further studies including peptide production and in vivo activity are in need.

## Figures and Tables

**Figure 1 antibiotics-11-00989-f001:**
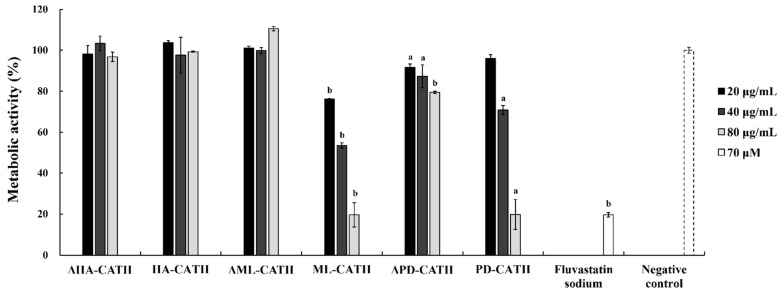
Metabolic activity after the addition of bat cathelicidin-derived peptides against *Mucor hiemalis* KCTC 26779. ΔHA-CATH, ΔML-CATH, and ΔPD-CATH are the predicted antimicrobial activity core region peptides of *Hipposideros armiger* (accession number, XP_019486615.1), *Myotis lucifugus* (accession number, XP_006108360.2), and *Phyllostomus discolor* (accession number, XP_028374415.1) cathelicidins, respectively. HA-CATH, ML-CATH, and PD-CATH are the predicted nascent mature region peptides. Statistical significance was indicated above bars; a (*p* < 0.01) and b (*p* < 0.001).

**Table 1 antibiotics-11-00989-t001:** Biochemical characteristics of bat cathelicidin-derived peptides.

Bat Cathelicidin-Derived Peptides	Sequence	Length	<H> ^a^	*z*^b^ (+)	Molecular Weight (Da)	Similarity		
						AMP ^c^	Source	(%) ^d^
ΔHA-CATH	LLRRGGRKIGQGLERIGQRIQGF	23	31	5	2609.08	SMAP-29	*Ovis aries*	46.87
HA-CATH	ILGRLRDLLRRGGRKIGQGLERIGQRIQGFFSNREPMEES	40	30	4	4640.37	K9CATH	*Canis familiaris*	51.22
ΔML-CATH	GIFILKHRRPIGRGIEIT	18	50	3	2076.52	Temporin-CPb	*Lithobates capito*	40
ML-CATH	LNPLIKAGIFILKHRRPIGRGIEITGRGIKKFFSK	35	40	8	3975.87	Palustrin-2CG1	*Amolops chunganensis*	38.64
ΔPD-CATH	IAGRIAGKLIGDAINRHRERNRQRR	25	36	6	2927.37	TP4	*Oreochromis niloticus*	44.83
PD-CATH	ILGPALRIGGRIAGRIAGKLIGDAINRHRERNRQRRG	37	35	8	4088.79	HKPLP	*Hippocampus kuda*	43.59

^a^ Ratio of hydrophobic residues. ^b^ Charge. ^c^ Most similar peptides among known antimicrobial peptides (AMPs). ^d^ Percentage of identical residues, including gaps, in the globally aligned sequence.

**Table 2 antibiotics-11-00989-t002:** Minimum inhibitory concentrations of three bat cathelicidin-derived peptides against various bacterial strains.

	Strain	Minimum Inhibitory Concentration (μg/mL, μM)					
		HA-CATH	ML-CATH	PD-CATH	chloramphenicol	Ampicillin	Gentamicin
Gram-negative bacteria	*Escherichia coli* ATCC 25922	18 (3.9)	2 (0.5)	7 (1.7)	3 (9.3)	5 (14.3)	1 (2.1)
	*Pseudomonas aeruginosa* ATCC 27853	>40 (8.6)	>40 (10.1)	>40 (9.8)	80 (247.6)	>640 (1831.7)	1 (2.1)
	*Salmonella enterica serovar* Typhimurium ATCC 14028	>40 (8.6)	21 (5.3)	22 (5.4)	5 (15.5)	>80 (228.8)	1 (2.1)
	*Acinetobacter baumannii* KCTC 23254	5 (1.1)	4 (1.0)	4 (1.0)	38 (117.6)	>80 (228.8)	10 (21.0)
	*Klebsiella pneumoniae* subsp. *pneumoniae* KCTC 1726	34 (7.3)	12 (3.0)	13 (3.2)	> 80 (247.6)	>80 (228.8)	19 (39.80
	*Enterobacter cloacae* subsp. *cloacae* ATCC 13047	>40 (8.6)	35 (8.8)	30 (7.3)	5 (15.5)	>80 (228.8)	1 (2.1)
Gram-positive bacteria	*Staphylococcus aureus* ATCC 6538	26 (5.6)	1 (0.3)	3 (0.7)	7.5 (23.2)	2 (5.7)	1 (2.1)
	*Bacillus cereus* ATCC 10876	25 (5.4)	3 (0.8)	6 (1.5)	10 (31.0)	80 (228.8)	1 (2.1)
	*Enterococcus faecalis* ATCC 29212	>40 (8.6)	5 (1.3)	6 (1.5)	10 (31.0)	10 (28.6)	90 (189.0)
	*Streptococcus agalactiae* ATCC 27956	>40 (8.6)	8 (2.0)	8 (2.0)	5 (15.5)	4 (11.4)	75 (157.5)
	*Streptococcus dysgalactiae* ATCC 27957	>40 (8.6)	14 (3.5)	18 (4.4)	4 (12.4)	2 (5.7)	15 (31.5)
	*Streptococcus equi* subsp. *zooepidemicus* ATCC 43079	>40 (8.6)	9 (2.3)	17 (4.2)	5 (15.5)	2 (5.7)	45 (94.5)

**Table 3 antibiotics-11-00989-t003:** Minimum inhibitory concentrations of six bat cathelicidin-derived peptides against two yeast form fungal strains.

Strain	MIC (μg/mL, μM)						
	ΔHA-CATH	HA-CATH	ΔML-CATH	ML-CATH	ΔPD-CATH	PD-CATH	Ciclopirox ^a^
*Candida albicans* KCTC 7270	>40 (15.3)	>40 (8.6)	>40 (19.3)	40 (10.1)	25 (8.5)	5 (1.2)	3.5 (16. 9)
*Cryptococcus cuniculi* KCTC 17232	>40 (15.3)	45 (9.7)	>40 (19.3)	5 (1.3)	15 (5.1)	5 (1.2)	0.5 (2.4)

^a^ Reference for antifungal activity.

**Table 4 antibiotics-11-00989-t004:** Viability of HaCaT cells after the addition of bat cathelicidin-derived peptides.

Treatment	Concentration (μg/mL)	Cell Viability ± SD (%)
ΔHA-CATH	64	100.5 ± 0.5
	160	97.7 ± 4.3
HA-CATH	64	100 ± 0.3
	160	97.5 ± 0.6
ΔML-CATH	64	100.6 ± 1.3
	160	96.0 ± 0.8
ML-CATH	64	99.7 ± 0.8
	160	97 ± 1.8
ΔPD-CATH	64	100.3 ± 0.3
	160	98.4 ± 2.3
PD-CATH	64	100.5 ± 0.2
	160	10.4 ± 0.4 *
Melittin	64	7.2 ± 0.03 *
Triton X-100	-	8.9 ± 0.05 *
Negative control	-	99.5 ± 0.83

Note: Statistical significance; * *p* < 0.001.

## Data Availability

The dataset analyzed during this study can be found in the NCBI database under the following accession numbers: XM_036219680.1, XM_019631070.1, XM_016202809.1, XM_006108300.3, XM_006108299.3, XM_006108298.3, XM_006108738.3, XM_033132622.1, XM_036030383.1, XM_028518614.2, XM_024566029.2, XM_024566030.2, XP_032960802.1, XP_032960796.1, XP_032960740.1, XP_032961053.1, XP_032991305.1, XP_032960777.1, XP_032960718.1, XP_032960717.1, XP_032959658.1, XP_032959919.1, XP_032959522.1, XP_032959179.1, XP_032952314.1, XP_032950360.1, XP_032951348.1, XP_032951380.1, XP_032951378.1, XP_032951381.1, XP_032951382.1, XP_032951383.1, XP_032951385.1, XP_032950352.1, XP_032959847.1, XP_032991304.1, XP_032988513.1, XP_032984445.1, XP_032952959.1, XP_036075573.1, KAF6473429.1, XP_019486615.1, XP_016058295.1, XP_006108362.1, XP_006108361.1, XP_006108360.2, XP_006108800.3, KAF6312595.1, KAF6312594.1, XP_035886276.1, XP_028374415.1, KAF6098810.1, KAF6098809.1, KAF6098808.1, KAF6100376.1, XP_024421797.1, and XP_024421798.1. Additional data supporting the conclusions of this study are available in the Appendix A.

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
