# Peer review of "Genomewide Analysis and Biological Characterization of Cathelicidins with Potent Antimicrobial Activity and Low Cytotoxicity from Three Bat Species"

_antibiotics, 2022, doi:10.3390/antibiotics11080989_

Round 1
Reviewer 1 Report
In this article, the authors conducted a genome-level analysis of the antimicrobial peptide cathelicidins from seven bat species and listed 19 cathelicidin-like sequences. In addition, three cathelicidins were synthesized chemically and evaluated for their biological functions along with cytotoxicity (HaCaT cells).
Although the article has scientific rigor, several major flaws need to be improved before publication.
Major comments:
[1] Title/Abstract: The title is too short. The abstract section is unsuitable—no focal point in the abstract section. Also, so many sentences are fragmented in the abstract section. Rewrite the methods, results, and conclusion (in the abstract) in a more straightforward form.
- I prefer Gram-positive, Gram-negative, in silico, etc. (not gram-positive, gram-negative, in-silico, etc.).
- Authors are suggested to use the full form when used for the first time throughout the manuscript.
- ‘The two cathelicidins also showed effective antifungal’ .....…Not clear.
- Delete I, we, our from the manuscript.
[2] Introduction: The introduction section is modest. Authors are suggested to develop the introduction section by adding the literature-related cathelicidin. Authors may add a reference with reference number 8 (https://doi.org/10.1080/07391102.2020.1850358).
- Why these seven bat species were selected?
- ‘Our results will contribute to a better understanding’…is this the future prospect?
[3] Results: The results section needs to be improved by adding significant results. Also, need to maintain a logical flow of the writing. Many grammatically problematic sentences are in the results section, which must be checked and corrected precisely.
- This inactivated serum (Table 4) does not make any sense to the data provided.
- The subtitles of the results can improve.
- ‘These results were consistent’ …not clear.
- ‘Therefore, our results suggest that the copy number’ ….this not fit here.
- ‘We were unable to observe the antibacterial activities of the short-form peptides’ …. Why is so?
[4] Discussion: It looks like their discussion is feeble and incomplete. As they done in silico studies to identify the peptides, I recommend exploring how they become good antibiotics. What’s the proposed mechanism of action? They avoid this in the discussion. Please, include the data from other sources about related works. They can do in silico analysis and discuss to wrap the story. Otherwise, it is incomplete and does not go as a research article. It could be published as a short communication.
- In the discussion, many concepts already reported in the introduction are repeated, so it is better to avoid unnecessary repetitions.
- ‘To the best of our knowledge, this is the first report of the antimicrobial activity of bat cathelicidins’ …..How do authors claim such?
[5] Materials and Methods: This section is written without proper references. Need a logical flow of the writings with enough references. Some of the methods need detailed description (e.g., In silico identification of cathelicidin-like sequences in the genome of seven bat species, Peptide synthesis). Only citing references is not enough. Access date must need to mention. The statistical analysis section is missing (what test, test details, significance, software, etc.).
- How are the synthesized cathelicidins (via solid-phase peptide synthesis) characterized/confirmed?
[6] Conclusion: It is missing. The conclusion needs to address future perspectives. The novelty of the work should be added by the author in the conclusion section.
English is modest. Therefore, the authors need to improve their writing style. In addition, the whole manuscript needs to be checked by native English speakers. Fundamental changes are obligatory before going for the final version.
Based on the above observations, I recommend a major revision of the manuscript antibiotics-1821787.
For your kind information, I was unable to check plagiarism.
Author Response
Cover letter
July 15, 2022
Dear Editor-in-Chief:
We wish to submit our revised manuscript, titled “Genomewide Analysis and Biological Characterization of Cathelicidins with Potent Antimicrobial Activity and Low Cytotoxicity from Three Bat Species,” coauthored by M. Choi and H. Cho et al., for publication in Antibiotics. We extensively modified our original submission throughout entire manuscript to accommodate reviewers’ comments. Changes were indicated in blue. The point-by-point responses from authors were described at the end of this letter.
We believe that this paper will be of interest to the readership of your journal because the characterization of novel antimicrobial peptides from bats may aid in the identification of novel antibacterial, antifungal, and antiviral therapeutic agents. Moreover, the three cathelicidins validated for their antimicrobial activities in this study can be used as references for the biological characterization of other bat antimicrobial peptides.
This manuscript has not been published or presented elsewhere in part or in its entirety and is not under consideration by another journal. We have read and understood your journal’s policies and believe that neither the manuscript nor the study violates any of these.
We look forward to your response.
Sincerely,
Chankyu Park
Department of Stem Cell and Regenerative Biotechnology
Konkuk University
Hwayang-dong, Gwangjin-gu
Seoul 05029, Republic of Korea
Tel: 82-2-450-3697
Fax: 82-2-457-8488
E-mail: chankyu@konkuk.ac.kr
Author response
Reviewer 1
[1] Title/Abstract: The title is too short. The abstract section is unsuitable—no focal point in the abstract section. Also, so many sentences are fragmented in the abstract section. Rewrite the methods, results, and conclusion (in the abstract) in a more straightforward form.
<Response> The title is changed to “Genomewide Analysis and Biological Characterization of Cathelicidins with Potent Antimicrobial Activity and Low Cytotoxicity from Three Bat Species”. The abstract was rewritten to meet the reviewer’s points. All the changes in the revision were indicated in blue.
Revised abstracts: Cathelicidins are potent antimicrobial peptides with broad spectrum antimicrobial activity in many vertebrates and an important component of the innate immune system. However, our understanding on the genetic variations and biological characteristics of bat cathelicidins is limited. In this study, we performed genome-level analysis of the antimicrobial peptide cathelicidins from seven bat species in the six families, listed 19 cathelicidin-like sequences, and showed that the number of functional cathelicidin genes differed among bat species. Based on the identified biochemical characteristics of bat cathelicidins, three cathelicidins, HA-CATH (from Hipposideros armiger), ML-CATH (from Myotis lucifugus), and PD-CATH (from Phyllostomus discolor), with clear antimicrobial signatures were chemically synthesized and evaluated antimicrobial activity. HA-CATH showed narrow-spectrum antibacterial activity against a panel of 12 reference bacteria, comprising 6 Gram-negative and 6 Gram-positive strains. However, ML-CATH and PD-CATH showed potent antibacterial activity against a broad spectrum of Gram-negative and Gram-positive bacteria with minimum inhibitory concentration (MIC) of 1 and 3 μg/mL, respectively, against Staphylococcus aureus. ML-CATH and PD-CATH also showed antifungal activities against Candida albicans and Cryptococcus cuniculi with MIC of 5 to 40 μg/mL, respectively, and 80% inhibition of the metabolism of Mucor hiemalis hyphae at 80 μg/mL, while displaying minimal cytotoxicity to HaCaT cells. Taken together, although the spectrum and efficacy of bat cathelicidins were species-dependent, the antimicrobial activity of ML-CATH and PD-CATH was comparable to that of other highly active cathelicidins in vertebrates while having negligible cytotoxicity to mammalian cells. ML-CATH and PD-CATH can be exploited as promising candidates for the development of antimicrobial therapeutics.
- I prefer Gram-positive, Gram-negative, in silico, etc. (not gram-positive, gram-negative, in-silico, etc.).
<Response> Changes were made across the manuscript.
- Authors are suggested to use the full form when used for the first time throughout the manuscript.
<Response> We checked possible errors and full forms were used for the first time of the use. HaCaT is the name of the cell line and we did not add any additional description.
- ‘The two cathelicidins also showed effective antifungal’ .....…Not clear.
<Response> The sentence was rewritten more clearly. See lines 25-26; “ML-CATH and PD-CATH also showed antifungal activities against...”
- Delete I, we, our from the manuscript.
<Response> We tried to remove the words except when we felt that it is necessary.
[2] Introduction: The introduction section is modest. Authors are suggested to develop the introduction section by adding the literature-related cathelicidin. Authors may add a reference with reference number 8 (https://doi.org/10.1080/07391102.2020.1850358).
<Response> We added a new paragraph regarding to the functions of cathelicidin related to our results. See lines 68-78 of the revised manuscript. “Recent studies have revealed various functions other than antibacterial activity of vertebrate cathelicidins, particularly for LL-37 and CRAMP of humans and mice, respectively [34, 35]. For example, a defective intestinal expression of CRAMP in nonobese diabetic (NOD) mouse induced the colonic dysbiosis and could contribute to autoimmunity in the pancreas in turn [36]. Also, endogenous CRAMP and LL-37 are suggested as inactivators of Zika virus by inhibiting viral replication and directly inactivating viral virions, thus exhibiting therapeutic potential against the viral infection in vivo [37]. The protective role of LL-37 against Candida auris known for causing severe infectious disease with high mortality attributed to multidrug-resistance against all major classes of antifungal drugs was also reported [38]. In addition, in vivo free radical scavenging ability of a cathelicidin was reported from frogs [39].” In addition, we added the reference suggested by the reviewer. See lines 46-47; “Currently, various efforts to develop therapeutics against SARS-CoV-2 are being made [11, 12].”
- Why these seven bat species were selected?
<Response> The reason was described in the revision. See lines 82-86; “In this study, we conducted genomewide in silico analysis of cathelicidin genes from seven bat species in the six families, Rousettus aegyptiacus, Hipposideros armiger, Miniopterus natalensis, Myotis lucifugus, Rhinolophus ferrumequinum, Phyllostomus discolor, and Desmodus rotundus, to maximize the genetic diversity”
- ‘Our results will contribute to a better understanding’…is this the future prospect?
<Response> Modified accordingly. See line 88 of the revised manuscript. “Our results contribute to a better understanding of the possible roles of AMPs…”
[3] Results: The results section needs to be improved by adding significant results. Also, need to maintain a logical flow of the writing. Many grammatically problematic sentences are in the results section, which must be checked and corrected precisely.
<Response> The several parts of results were rewritten or modified to accommodate reviewer’s points. The changes were indicated in blue. See lines 98-101, 123-125, 132-148, 152-154, 175-182, 189-207, and 228-234;
“using National Center for Biotechnology Information (NCBI) genome blast with query coverage > 50% and e-value < 0.001 (Table S1). The number of AMP genes identified from our analysis was consistent with the gene numbers reported in R. ferrumequinum for each AMP family in the NCBI protein database without finding additional AMP genes.”,
“cathelicidin-like sequences (XP_024421797.1 and XP_024421798.1). Similarly, two cathelicidin-like sequences, KAF6312594.1 and XP_032988513.1, were encoded from an identical locus in R. ferrumequinum, and the structure of KAF6312595.1 lacks an exon.”,
“We conducted an in silico prediction of the antimicrobial activity core region for the 19 bat cathelicidin-like sequences to define the signal peptide, conserved CLD domain, and mature peptide region using two different prediction tools, antimicrobial sequence scanning system (AMPA) and database of antimicrobial activity and structure of peptides (DBAASP) (Table S4). Consequently, five putatively functional sequences, KAF6473429.1, XP_019486615.1, XP_006108360.2, XP_035886276.1, and XP_028374415.1, with the complete pre-pro protein structure of cathelicidins and predicted presence of core antimicrobial activity regions were determined (Table S5). Among them, three predicted antimicrobial activity core regions corresponding to the cathelicidins of H. armiger (XP_019486615.1), M. lucifugus (XP_006108360.2) and P. discolor (XP_028374415.1) which were named as ΔHA-CATH, ΔML-CATH, and ΔPD-CATH where “Δ” indicate a deletion of several amino acids from the nascent mature peptide region, namely HA-CATH, ML-CATH, and PD-CATH, respectively, were selected and subjected to the analysis of biochemical features using Antimicrobial Peptide Database (APD3). The three antimicrobial activity core regions or short form peptide regions (ΔHA-CATH, ΔML-CATH, and ΔPD-CATH) are consisted of 18 to 25 amino acids with 2.08 to 2.93 kDa in molecular weight,”,
“Their amino acid sequence similarity to known cathelicidins was highest with cathelicidins from sheep (SMAP-29), dogs (K9CATH), frogs (Temporin-CPb, Palustrin-2CG1), and fish (TP4, HKPLP) and the level ranged from 38.64–51.22%.”
“Gram-negative strains and S. aureus and B. cereus among Gram-positive strains, indicating that the antimicrobial potency and spectrum of bat cathelicidins varied among species. However, ΔHA-CATH, ΔML-CATH, and ΔPD-CATH, the short-form peptides corresponding to the predicted antibacterial core region, did not show any observable antibacterial activity against all strains of the reference bacterial panel, even at a high concentration (> 40 μg/mL or 13.7 μM), indicating the prediction of active core region for the bat cathelicidins was not accurate.”
“Six bat-cathelicidin peptides (ΔHA-CATH, ΔML-CATH, ΔPD-CATH, HA-CATH, ML-CATH, and PD-CATH) were evaluated for the presence of antifungal activity against three different fungal species, Candida albicans (C. albicans), Cryptococcus cuniculi (C. cuniculi), and Mucor hiemalis (M. hiemalis) (Table 3; Figure 1). The antifungal effect of the yeast-type fungi, C. albicans and C. cuniculi, was estimated in a manner similar to the antibacterial activity method. However, the antifungal activity of M. hiemalis, a mold-type fungus, was estimated based on the inhibition of the metabolic rate against hyphae. The treatment of ML-CATH and PD-CATH showed effective antifungal activities against the two yeast forms fungi with MIC of 5 to 40 μg/mL (1.2 to 10.1 μM) and M. hiemalis with a similar rate of metabolic inhibition (~80%) to fluvastatin sodium. In addition, intermediate levels of antifungal activity were observed against C. albicans and C. cuniculi from the treatment of ΔPD-CATH which did not show antibacterial activity against our reference bacterial panel, indicating the inherently strong antimicrobial potency of PD-CATH compared to the other bat cathelicidins tested in this study and differences in the effect of ΔPD-CATH against bacterial and fungal cell walls. In addition, HA-CATH only showed slight antifungal activity (MIC=45 μg/mL or 9.7 μM) against C. cuniculi, indicating limited activity against fungi which is consistent to the antibacterial activity of HA-CATH (Table 2). The antifungal activity of the tested peptides seems to be higher for yeast than mold-type fungi.”
“showing significant cytotoxicity at the high concentration. Therefore, the cytotoxicity on mammalian cells showed a positive correlation with the potency of bat cathelicidins. In addition, melittin, an AMP known to have a potent cell lytic activity, showed severe cell lysis, with 7.2% cell viability at 64 μg/mL concentration. These results indicate that the cytotoxic effect of HA-CATH and ML-CATH is at a minimal level and given the effective bactericidal concentration of PD-CATH (3-30 μg/mL), the cytotoxicity is within the acceptable range for therapeutic use.”
- This inactivated serum (Table 4) does not make any sense to the data provided.
<Response> Removed.
- The subtitles of the results can improve.
<Response> Subtitles were improved. See lines 103-104, 162 and 188 of the revision,
“2.2. Differences in the number of functional cathelicidin genes among seven bat species belonging to different families”,
“2.4. Confirmation of the antibacterial activity of three bat cathelicidins”,
“2.5. Confirmation of the antifungal activity of three bat cathelicidins”
- ‘These results were consistent’ …not clear.
<Response> The sentence was rewritten. See lines 99-101: “The number of AMP genes identified from our analysis was consistent with the gene numbers reported in R. ferrumequinum for each AMP family in the NCBI protein database without finding additional AMP genes.”
- ‘Therefore, our results suggest that the copy number’ ….this not fit here.
<Response> We moved the sentences to the discussion section. See lines 334-338; “Therefore, the copy number of cathelicidins in bats could vary from a single copy to multiple copies depending on the species, which is consistent with a previous analysis [43]. However, further refinement of the current bat genome assemblies and annotations is needed to determine the copy number of cathelicidin family genes more accurately.”
- ‘We were unable to observe the antibacterial activities of the short-form peptides’ …. Why is so?
<Response> The sentence was rewritten and the details were also described in the discussion. See lines 178-182: “However, ΔHA-CATH, ΔML-CATH, and ΔPD-CATH, the short-form peptides corresponding to the predicted antibacterial core region, did not show any observable antibacterial activity against all strains of the reference bacterial panel, even at a high concentration (> 40 μg/mL or 13.7 μM), indicating the prediction of active core region for the bat cathelicidins was not accurate.”
[4] Discussion: It looks like their discussion is feeble and incomplete. As they done in silico studies to identify the peptides, I recommend exploring how they become good antibiotics. What’s the proposed mechanism of action? They avoid this in the discussion. Please, include the data from other sources about related works. They can do in silico analysis and discuss to wrap the story. Otherwise, it is incomplete and does not go as a research article. It could be published as a short communication.
<Response> The recommended points were included in the revision. See lines 251-258, 259-268, 295-317 and 334-338:
“Although few studies reported bioinformatic prediction of bat cathelicidins based on available sequencing information and experimental characterization of their antimicrobial activity [43, 50], the number of tested bacterial strains and peptides was highly limited, and other characteristics such as fungicidal and metabolically inhibitory activities against fungi and cytotoxicity have not been validated experimentally. To the best of our knowledge, this is the first report of the antimicrobial activity of bat cathelicidins against diverse bacteria and fungi associated with human health and food pathogens.”,
“A previous study briefly reported the bactericidal effect of a bat cathelicidin, Ml-LN-35, which is identical to ML-CATH, against E. coli and B. cereus [50]. The results were consistent to the results in this study with MICs ranging from 0.5 to 0.8 μM (Table 2). In contrast, antibacterial activity against S. aureus ATCC 6538 was not detected in the previous study, while the peptide showed highly potent anti-staphylococcal activity (1 μg/mL or 0.3 μM) as shown in Table 2. This inconsistency might be attributed to different experimental condition including media composition and peptide synthesis with some modification such as acetylation and amidation in N- and C-terminal end, respectively, which can affect the nature and structure of AMPs and influence biological activity and stability of the molecules in turn [51-53].”,
“In addition to the extensive functional studies on LL-37 and CRAMP, recent interests in AMPs as potent antibiotic agents lead to the identification and characterization of novel cathelicidins from diverse species including Python bivittatus (Burmese python), Heterocephalus glaber (naked mole-rat), Monodelphis domestica (gray short-tailed opossum), Lates calcarifer (Asian sea bass) and Sarcophilus harrisii (Tasmanian devil) [25, 47, 54, 59, 60]. These studies suggested that cathelicidins are promising candidates to develop antimicrobial therapeutics considering potent broad-spectrum bactericidal activity, minimal cytotoxicity to mammalian cells, and inducibility of lower antibiotic resistance comparing to classical antibiotics. Interestingly, bat cathelicidins, ML-CATH and PD-CATH, showed much broader and stronger antibacterial activities than all the peptides described above (Table 2). Furthermore, ML-CATH and PD-CATH showed milder toxicity comparing to ModoCath1 which is one of the cathelicidins showing broad spectrum with high potency. The viability of human keratinocyte was affected at a two-fold higher concentration (64 µg/mL) in ML-CATH and PD-CATH comparing to ModoCath1 (32 µg/mL) (Table 4). These characteristics could make ML-CATH and PD-CATH more suitable for pharmaceutical applications than other cathelicidins. However, further studies are still required to overcome intrinsic disadvantages of AMPs including high production cost, low stability in vivo, and deficiency of proper delivery system. An effective production system of recombinant cathelicidins using an engineered green fluorescent protein (GFP) as a tag protein was attempted for large scale production of AMPs [61]. The conjugation of AMPs with nanoparticles has been suggested to overcome poor stability of AMPs in biological fluids and proteolytic degradation and to control pharmacokinetics [62-64].”,
“Therefore, the copy number of cathelicidins in bats could vary from a single copy to multiple copies depending on the species, which is consistent with a previous analysis [43]. However, further refinement of the current bat genome assemblies and annotations is needed to determine the copy number of cathelicidin family genes more accurately.”
- In the discussion, many concepts already reported in the introduction are repeated, so it is better to avoid unnecessary repetitions.
<Response> We improved it across the discussion and many changes were made as shown above.
- ‘To the best of our knowledge, this is the first report of the antimicrobial activity of bat cathelicidins’ …..How do authors claim such?
<Response> We rephased the parts. See lines 251-258;
“Although few studies reported bioinformatic prediction of bat cathelicidins based on available sequencing information and experimental characterization of their antimicrobial activity [43, 50], the number of tested bacterial strains and peptides was highly limited, and other characteristics such as fungicidal and metabolically inhibitory activities against fungi and cytotoxicity have not been validated experimentally. To the best of our knowledge, this is the first report of the antimicrobial activity of bat cathelicidins against diverse bacteria and fungi associated with human health and food pathogens.”
[5] Materials and Methods: This section is written without proper references. Need a logical flow of the writings with enough references. Some of the methods need detailed description (e.g., In silico identification of cathelicidin-like sequences in the genome of seven bat species, Peptide synthesis). Only citing references is not enough. Access date must need to mention. The statistical analysis section is missing (what test, test details, significance, software, etc.).
- How are the synthesized cathelicidins (via solid-phase peptide synthesis) characterized/confirmed?
<Response> We improved M&M by adding additional details, references, and a supplementary table (Table S7). The statistical analysis section was added and we also included the statistical significance of the cell viability and metabolic activity in the revision. (Table 4, Figure 1). The final access dates of public databases were indicated. See lines 355-357, 357-359, 359-361, 375-376, 381-384, 391-394, 402-403, 438-442 and 491-493;
“Four hundred sequences belonging to five major AMP families of vertebrates, alpha-defensins, beta-defensins, cathelicidin, hepcidin, and LEAP-2, were obtained similarly to the previous studies [25, 47].”
“Briefly, the sequences of 2,477 non-redundant AMPs were downloaded from UniProtKB/Swiss-Prot (http://www.uniprot.org/uniprot/; [73]) using the query “antimicrobial peptide AND reviewed: yes”.”,
“After removing sequences originated from non-vertebrates, the sequences corresponding to five major AMP families described above were collected (Table S7) [25].”,
“The last accession of genome databases was May 1 of 2022.”,
“AMPA (https://tcoffee.crg.eu/apps/ampa/do; [76]) and DBAASP (https://dbaasp.org/; [77]) were used to predict the potential antimicrobial activity domains using the default window size and threshold value.”,
“Peptides corresponding to the predicted active core and mature peptide regions of cathelicidins were generated via solid-phase peptide synthesis, purified via high-performance liquid chromatography (HPLC), and confirmed by mass spectroscopy using a commercial service (GenScript, Piscataway Township, NJ, USA).”,
“The synthesized peptides were dissolved in distilled water at a concentration of 4 mg/mL.”,
“Cells were prepared in YMB for C. albicans and RPMI 1640 medium (with l-glutamine, without phenol red and sodium bicarbonate, Sigma Aldrich) and adjusted to pH 7.0 with 0.165 M 3-(N-morpholino) propanesulfonic acid (MOPS) (iNtRON Biotechnology, Seoul, Korea), as previously described [80] for C. cuniculi.”
“ 4.7. Statistical analysis. Statistical analysis was performed using Student's paired t-test with a two-tailed distribution using RStudio (https://www.R-project.org/).”
[6] Conclusion: It is missing. The conclusion needs to address future perspectives. The novelty of the work should be added by the author in the conclusion section.
<Response> We included the conclusion section in the revision. See lines 495-507;
“5. Conclusions. In summary, species-specific differences in genetic and functional characteristics of cathelicidins from seven bat species were studied. We listed 19 cathelicidin-like sequences from seven bat species and tested the biological activity of three cathelicidins (HA-CATH, ML-CATH, and PD-CATH). Among them, ML-CATH and PD-CATH showed potent antibacterial activity against a broad spectrum of Gram-negative and Gram-positive bacteria. The potency of antibacterial activity of ML-CATH and PD-CATH was comparable to that of other highly active cathelicidins reported in vertebrates. ML-CATH and PD-CATH also showed antifungal activities against C. albicans, C. cuniculi, and M. hiemalis. The cytotoxicity on HaCaT cells showed a positive correlation with the potency of bat cathelicidins, but given the effective bactericidal concentration, the cytotoxicity of bat cathelicidins is within the acceptable range for therapeutic exploitation. ML-CATH and PD-CATH are promising candidates for the development of peptide antibiotics, but further studies including peptide production and in vivo activity are in need.”
Authors greatly appreciate reviewer’s valuable comments.
Please see the attachment

Reviewer 2 Report
Authors identified antimicrobial peptide cathelicidins from the bats-HA-CATH (from Hipposideros armiger), ML-CATH (from Myotis lucifugus) and PD-CATH (from Phyllostomus discolor).
1. My concern is in-silico findings can be proved with minimum experimentation and resubmit the ms for authenticity of this findings.
2. Results of the cell viability should be subjected to appropriate statistical tests for consider this ms for publication.
3. Antibacterial activity was tested with 6 each gram positive and negative bacterium, however, anti-fungal activity tested with only 2 fungi. This should be increased at least by 6 organisms
Author Response
Cover letter
July 15, 2022
Dear Editor-in-Chief:
We wish to submit our revised manuscript, titled “Genomewide Analysis and Biological Characterization of Cathelicidins with Potent Antimicrobial Activity and Low Cytotoxicity from Three Bat Species,” coauthored by M. Choi and H. Cho et al., for publication in Antibiotics. We extensively modified our original submission throughout entire manuscript to accommodate reviewers’ comments. Changes were indicated in blue. The point-by-point responses from authors were described at the end of this letter.
We believe that this paper will be of interest to the readership of your journal because the characterization of novel antimicrobial peptides from bats may aid in the identification of novel antibacterial, antifungal, and antiviral therapeutic agents. Moreover, the three cathelicidins validated for their antimicrobial activities in this study can be used as references for the biological characterization of other bat antimicrobial peptides.
This manuscript has not been published or presented elsewhere in part or in its entirety and is not under consideration by another journal. We have read and understood your journal’s policies and believe that neither the manuscript nor the study violates any of these.
We look forward to your response.
Sincerely,
Chankyu Park
Department of Stem Cell and Regenerative Biotechnology
Konkuk University
Hwayang-dong, Gwangjin-gu
Seoul 05029, Republic of Korea
Tel: 82-2-450-3697
Fax: 82-2-457-8488
E-mail: chankyu@konkuk.ac.kr
Author response
Reviewer 2
- My concern is in-silico findings can be proved with minimum experimentation and resubmit the ms for authenticity of this findings.
<Response> Extensive in silico analysis of bat cathelicidins were reported recently by de la Lasta et al.(2021, Bioinformatic analysis of genome-predicted bat cathelicidins). Although the contents of study are completely different between the two studies, in silico analysis of bat cathelicidins were extensively covered by de la Lasta et al. Therefore, we limited our in silico analysis to avoid redundancy. Please bear with such circumstances.
- Results of the cell viability should be subjected to appropriate statistical tests for consider this ms for publication.
<Response> We included the statistical significance of the cell viability and metabolic activity in the revision. See Table 4. Figure 1, and lines 491-493;
“4.7. Statistical analysis. Statistical analysis was performed using Student's paired t-test with a two-tailed distribution using RStudio (https://www.R-project.org/).”
- Antibacterial activity was tested with 6 each gram positive and negative bacterium, however, anti-fungal activity tested with only 2 fungi. This should be increased at least by 6 organisms
<Response> Antifungal activity was tested against 3 different strains of fungi, 2 yeast type and 1 mold type. In contrast to bacteria, the analysis of MIC for fungi are much more complex and, as the reviewer are aware, the MIC assay using AMPs against fungi are still not well established because of difficulties in dealing with complications associated with culture condition, proliferation rate, and methodological biases which are unique for each strain. Therefore, we were able to obtain reliable results from only three fungi for this study. Further studies will be followed.
Authors greatly appreciate reviewer’s valuable comments.
Please see the attachment

Reviewer 3 Report
The manuscript, Determination and Biological Characterization of Cathelicidin- Derived Peptides from Three Bat Species, describes the anti-bacterial, anti-fungal, and mammalian cytotoxic activities of a select number of anti-microbial peptides (AMPs) derived from several species of bat. Citing previous work on AMPs and their various implications on the mammalian immune response, the authors suggest a possible corollary of their findings to the potential use of cathelicidin AMPs as anti-viral therapeutics. The significance of this research is timely, as bats have been shown to be asymptomatic carriers of various viruses (including coronavirus) that adversely affect other mammalian populations. Therefore, gaining a greater understanding of how cathelicidin AMPs benefit the immune responses of bats may be beneficial in creating antimicrobial therapeutics (in part to presumably to mitigate the effects of viral infection).
Overall, I find the manuscript to be of high quality. The authors provide a logical pipeline of experiments to characterize the AMPs of interest, beginning with genome wide in silico analysis that substantiates their focus on canthelicidins. I appreciate the characterization of sequences according to their hydrophobicity index and net charge, as these are vital to the function of AMPs in general. The results of the ensuing anti-bacterial and anti-fungal studies are illuminating, in showing that core AMP sequences (the delta series) are generally not enough to elicit anti-bacterial/anti-fungal activity, and that at least two of the peptides show negligible cytotoxic effects on human cells. In the discussion, the authors make the case for a structure-activity relationship that arises from deleting key residues that may impact helix formation of the peptides, thus also affecting anti-microbial activity. While this manuscript doesn’t dive into that possibility by experiment, the authors suggest a follow-up that examines the importance of key residues and examining the SAR component.
My suggestions for further improvement are very minor:
In section 2.4, the MIC values are stated in both microgram/mL followed by micromolar in parentheses. Later sections omit the micromolar values. For narrative consistency, state the MIC values in parentheses. The ensuing tables very nicely lay this information out.
I assume that the peptides have a free amine and carboxylate at the N- and C- termini, respectively. If this is the case, no action needed. However, if you implemented any caps (which is common in some studies) please state as much. Also, when describing the antimicrobial assays, please briefly note how the peptide stock concentrations were prepared.
Author Response
Cover letter
July 15, 2022
Dear Editor-in-Chief:
We wish to submit our revised manuscript, titled “Genomewide Analysis and Biological Characterization of Cathelicidins with Potent Antimicrobial Activity and Low Cytotoxicity from Three Bat Species,” coauthored by M. Choi and H. Cho et al., for publication in Antibiotics. We extensively modified our original submission throughout entire manuscript to accommodate reviewers’ comments. Changes were indicated in blue. The point-by-point responses from authors were described at the end of this letter.
We believe that this paper will be of interest to the readership of your journal because the characterization of novel antimicrobial peptides from bats may aid in the identification of novel antibacterial, antifungal, and antiviral therapeutic agents. Moreover, the three cathelicidins validated for their antimicrobial activities in this study can be used as references for the biological characterization of other bat antimicrobial peptides.
This manuscript has not been published or presented elsewhere in part or in its entirety and is not under consideration by another journal. We have read and understood your journal’s policies and believe that neither the manuscript nor the study violates any of these.
We look forward to your response.
Sincerely,
Chankyu Park
Department of Stem Cell and Regenerative Biotechnology
Konkuk University
Hwayang-dong, Gwangjin-gu
Seoul 05029, Republic of Korea
Tel: 82-2-450-3697
Fax: 82-2-457-8488
E-mail: chankyu@konkuk.ac.kr
Author response
Reviewer 3
- In section 2.4, the MIC values are stated in both microgram/mL followed by micromolar in parentheses. Later sections omit the micromolar values. For narrative consistency, state the MIC values in parentheses. The ensuing tables very nicely lay this information out.
<Response> We included uM values in parentheses. See lines 174, 181, 197, 204, 426 and 444
- I assume that the peptides have a free amine and carboxylate at the N- and C- termini, respectively. If this is the case, no action needed. However, if you implemented any caps (which is common in some studies) please state as much. Also, when describing the antimicrobial assays, please briefly note how the peptide stock concentrations were prepared.
<Response> No modifications were added for the peptides used in this study. We also included a sentence for a concentration of peptide stock. See lines 402-403;
“The synthesized peptides were dissolved in distilled water at a concentration of 4 mg/mL.”
Authors greatly appreciate reviewer’s valuable comments.
Please see the attachment.

Round 2
Reviewer 1 Report
I have gone through the manuscript in all the sections. I am happy that the authors improved the manuscript based on my suggestions. However, (a) authors need to use an appropriate preposition in the whole manuscript (e.g. ‘understanding on’ should be ‘understanding of’; line 14); and (b) the presentation of references should be uniform (journal name). I believe that the authors will correct these minor changes now (also, during the next stages publication process).
Author Response
Author response
Reviewer 1
I have gone through the manuscript in all the sections. I am happy that the authors improved the manuscript based on my suggestions. However, (a) authors need to use an appropriate preposition in the whole manuscript (e.g. ‘understanding on’ should be ‘understanding of’; line 14); and (b) the presentation of references should be uniform (journal name). I believe that the authors will correct these minor changes now (also, during the next stages publication process).
<Response>
(a) Based on the reviewer’s advice, several prepositions have been modified across the manuscript. All the changes were indicated in red. See lines 14, 266, 307-308, 523, 530
(b) We also corrected the format of references in a uniform way.
Authors greatly appreciate reviewer’s valuable comments.
Please see the attachment.

Reviewer 2 Report
Kindly clarify the following doubts
1. Figure 1. *P<0.01 and **P<0.001 can be replaced with a (*P<0.01) and b (P<0.001)
2. Table 3. I find only two fungi namely Candida albicans and Cryptococcus cuniculi. But author in the response, it is stated that 2 yeast type and 1 mold type. Kindly check and ratify my doubt
3. Table 3, author included MIC, but in the response it was specified that the analysis of MIC for fungi are much more complex, MIC assay using AMPs against fungi are still not well established because of difficulties in dealing with complications associated with culture condition, proliferation rate, and methodological biases which are unique for each strain. I agreed. Since MIC was recorded for 2 fungi, as I suggested i could also extended to at least for two fungi to get clear idea about anti-fungal activity
Author Response
Author response
Reviewer 2
- Figure 1. *P<0.01 and **P<0.001 can be replaced with a (*P<0.01) and b (P<0.001)
<Response>
We changed the indication of statistical significance from asterisk to letter (a and b). See Figure 1 and lines 219-220; “Statistical significance was indicated above bars; a (P<0.01) and b (P<0.001).”
- Table 3. I find only two fungi namely Candida albicansand Cryptococcus cuniculi. But author in the response, it is stated that 2 yeast type and 1 mold type. Kindly check and ratify my doubt
<Response>
To evaluate antifungal activities of bat cathelicidins, we used totally three fungi strains, consisting of two yeast fungi Candida albicans and Cryptococcus cuniculi, and one mold fungus, Mucor hiemalis. See section 4.5, Table 3 and Figure 1.
- Table 3, author included MIC, but in the response it was specified that the analysis of MIC for fungi are much more complex, MIC assay using AMPs against fungi are still not well established because of difficulties in dealing with complications associated with culture condition, proliferation rate, and methodological biases which are unique for each strain. I agreed. Since MIC was recorded for 2 fungi, as I suggested i could also extended to at least for two fungi to get clear idea about anti-fungal activity
<Response>
According to the comment in previous revision, it was mentioned that “3. Antibacterial activity was tested with 6 each gram positive and negative bacterium, however, anti-fungal activity tested with only 2 fungi. This should be increased at least by 6 organisms”. Thus, in our response previously, we asked to be excused that we could not extend the number of fungi to evaluate the antifungal activity of bat cathelicidins. Since we used three fungi in this study (please check section 4.5, Table 3 and Figure 1), it could be enough to get clear idea for antifungal activity, as you mentioned above.
Authors greatly appreciate reviewer’s valuable comments.
Please see the attachment.
